# The Stress of Fungicides Changes the Expression of Clock Protein CmFRQ and the Morphology of Fruiting Bodies of *Cordyceps militaris*

**DOI:** 10.3390/jof10020150

**Published:** 2024-02-13

**Authors:** Jing-Mei Peng, Dan-Dan Zhang, Zi-Yan Huang, Ming-Jia Fu

**Affiliations:** College of Life Science, Jiangxi Normal University, No. 99, Ziyang Avenue, Nanchang 330022, China; ade13929784412@163.com (J.-M.P.); danbaobaoeat@126.com (D.-D.Z.); ziyhuang@126.com (Z.-Y.H.)

**Keywords:** *Cordyceps militaris*, stress of fungicides, clock protein CmFRQ, degeneration and rejuvenation of fruiting body

## Abstract

The physiological, biochemical, and morphological changes brought about by fungi in response to fungicides can undoubtedly bring diversity to fungi. *Cordyceps militaris* strains TN (mating type genes *MAT1-1-1*, *MAT1-1-2*, and *MAT1-2-1*) and CmFRQ-454 (mating type genes *MAT1-1-1* and *MAT1-1-2*) were treated with non-lethal doses of fungicides amphotericin B, L-cysteine, terbinafine, and 5-fluorocytosine. The results showed that the treatment with amphotericin B, terbinafine, and 5-fluorocytosine promoted an increase in the relative content of clock protein CmFRQ (*C. militaris* FREQUENCY) in the mycelium of strain TN, while the high concentration of L-cysteine inhibited the expression of CmFRQ in strain TN. These four fungicides could reduce the relative contents of CmFRQ in the mycelium of strain CmFRQ454. The relative contents of CmFRQ in the mycelium of strain TN were increased after removing the four fungicides, but the relative contents of CmFRQ in the mycelium of strain CmFRQ454 were decreased after removing the four fungicides. This indicates that the effect of fungicides on CmFRQ on mycelium was still sustained after removing the stress of fungicides, and the operation of the circadian clock was changed. The fruiting bodies of *C. militaris* strain TN and CmFRQ-454 were still degenerated to varying degrees after removing amphotericin B, L-cysteine, and terbinafine. However, the fruiting bodies of strain TN after removing 5-fluorocytosine did not show significant degeneration; the fruiting bodies of strain CmFRQ-454 after removing 5-fluorocytosine obtained rejuvenation. These results indicate that the stress of fungicides could lead to the degeneration of fruiting bodies as well as the rejuvenation of fruiting bodies, resulting in the morphological diversity of *C. militaris*. The increase or decrease of the CmFRQ-454, the main component of the circadian clock, caused by the stress of fungicants, might lead to the differential degeneration of different mating-type strains of *C. militaris*.

## 1. Introduction

The effects of fungicides on fungi are varied, especially on the diversity of fungi. The use of fungicides creates a strong selection pressure on the fungus, which reduces the effectiveness of the fungicide. Scanlan et al. showed that the DNA insertions into the promoter of the erg11/CYP51 DMI target gene resulted in changes in the canola pathogen *Leptosphaeria maculans* resistance to demethylation inhibitor (DMI) fungicides [1]. Some fungicides induce a genetic change in *Monilinia fructicola* in the form of transposon transposition [2]. The genomic instability (single nucleotide polymorphisms, insertions/deletions, copy number variants, and transposable element insertions) in fungal plant pathogens caused by exposure to sublethal fungicide doses accelerates the emergence of fungicide resistance or other adaptive traits [3]. Many studies have been reported on the genetic diversity of fungi driven by fungicides, although there are differences in the characteristics that appear [4,5,6,7]. In addition, antifungal resistance caused by the use of fungicides has also received more attention [8,9,10].

Circadian clock mechanisms exist in many eukaryotic cells to regulate morphology, physiology, biochemistry, growth, and development [11,12,13]. The changes in the external environment can lead to a reset of the circadian clock to maintain synchronization with environmental conditions [14,15,16]. A typical circadian clock consists of three parts: environmental signal input, circadian oscillator, and time information output, which regulate a variety of different molecular and physiological processes and show circadian rhythm [17]. In *Neurospora crassa*, the key components of the core circadian oscillator include clock protein frequency (FRQ), blue light receptor white collar-1 (WC-1) and white collar-2 (WC-2), etc., and participate in the transcription-/translation-based negative feedback loop [18,19]. WC-1 and WC-2 can form a dimer (WCC) that acts as a photoreceptor, as well as a transcription factor or a positive element promoting *frq* transcription in the dark [20,21,22,23,24]. FRQ and FRH (FRQ-interacting RNA helicase) form an FRQ-FRH complex (FFC) functioning as the negative limb in the negative feedback loop [25,26,27]. After phosphorylation of FRQ by several kinases (CKI and CKII), the FRQ ubiquitination leads to its degradation, releasing WCC dimers, reactivating FRQ expression, and restarting the circadian cycle [14,28,29,30,31,32,33,34,35,36,37].

As a powerful external factor that interferes with fungi, fungicides undoubtedly bring about genetic diversity in fungi, and thus affect the physiological and biochemical characteristics of fungi, thereby affecting their morphology. In this study, we selected four fungal inhibitors: amphotericin B, L-cysteine, terbinafine, and 5-fluorocytosine. Amphotericin B is a polyene antifungal drug whose main mechanism of action is to damage the permeability of the cell membrane by combining with steroids on the cell membrane of sensitive fungi, causing the leakage of important substances in the cell, thus destroying the normal metabolism of the cell and inhibiting the growth of the fungus. Amphotericin B has the advantages of high antifungal activity, a broad antifungal spectrum, and low drug resistance [38]. Amino acids are important components of proteins, but a certain concentration of amino acids has an inhibitory effect on fungi [39,40,41]. Previous studies have shown that a certain concentration of L-cysteine can inhibit the growth of the mycelium of *Villosiclava virens* and inhibit the growth of the mycelium, spore germination, and germ tube elongation of *Alternaria alternata* [42,43]. Obviously, amino acid antifungal agents have good safety in food applications. Terbinafine is an allylamine broad-spectrum antifungal agent that inhibits squalene cyclooxygenase during the synthesis of ergosterol in fungal cells and accumulates squalene in cells, thus exerting a fungicidal effect. Previous studies have shown that terbinafine has antifungal effects on various fungi [44]. Under the action of fungal cytosine deaminase, 5-fluorocytosine enters fungal cells and is converted into fluorouracil, replacing uracil in RNA, thereby interfering with the synthesis of normal fungal proteins and achieving antifungal effects [45,46]. It can act on yeast (*Saccharomyces cerevisiae*), *Aspergillus*, *dematiaceous fungi*, *Candida albicans*, and *Cryptococcus neoformans* [47,48,49].

*Cordyceps militaris* is an economically important edible and medicinal fungus in China. With the increase in the number of subcultures, the fungal fruiting body is prone to degeneration, which affects its economic value [50]. Many causes of *C. militaris* degeneration have been analyzed, including environmental causes [50]. Circadian clocks can be directly influenced by the environment, and the FRQ of fungal circadian clocks is a centric component [17,51]. Therefore, *C. militaris* frequency (CmFRQ) was selected as the research object in this study.

## 2. Materials and Methods

### 2.1. C. Militaris Strains

*C. militaris* strain TN had mating-type genes *MAT1-1-1*, *MAT1-1-2*, and *MAT1-2-1*. *C. militaris* strain CmFRQ454 had mating-type genes *MAT1-1-1* and *MAT1-1-2*, but no *MAT1-2-1*. Strain CmFRQ-454 had been degenerated in the process of subculture.

### 2.2. Strain Activation

The *C. militaris* strains were activated before use. The strains were inoculated on potato dextrose agar (PDA; 200 g potatoes, 1 g KH_2_PO_4_, 1 g MgSO_4_·7H_2_0, 10 mg vitamin B1, 20 g glucose, 20 g agar powder, the fixed volume of 1000 mL after preparation) medium and cultured under constant temperature and darkness at 25 °C for 6 d. The obtained strains can be used in subsequent experiments.

### 2.3. Culture of Mycelium with the Treatment of Fungicides

*C. militaris* mycelium was cultured in PDA medium at 25 °C under dark conditions. When inoculating, 4-point holes were drilled on the plate of the PDA medium with a hole puncher (diameter 6 mm), and then the strain obtained in the same way was filled into the holes of the PDA medium. If the treatment of fungicides was required in the mycelium growth process, fungicides should be added to the cooled but unsolidified PDA medium according to the required concentration when preparing the PDA medium. The final concentrations of the fungicides added were 2 μg/mL for amphotericin B, 125 μg/mL for L-cysteine, 0.5 μg/mL for terbinafine, and 4.65 × 10^−6^ nmol/L for 5-fluorocytosine. At the same time, a blank control was set. The subculture mycelium of *C. militaris* obtained by the treatment of fungicides was S_0_. The S_0_ colony morphology was observed on the 5th and 10th days of culture. All S_0_ samples, together with Petri dishes, were wrapped in aluminum foil and placed in a –55 °C refrigerator to facilitate simultaneous protein extraction.

### 2.4. Transfer Culture of C. Militaris Mycelium after the Treatment of Fungicides

After the subculture S_0_ samples were treated with fungicides, the S_0_ mycelium of *C. militaris* was transferred onto the PDA without fungicides and then cultured at 25 °C under dark conditions to obtain the subculture S_1_. All S_1_ samples, together with Petri dishes, were wrapped in aluminum foil and placed in a –55 °C refrigerator to facilitate simultaneous protein extraction.

### 2.5. Culture of Fruiting Bodies for S_1_

*C. militaris* subculture S_1_ was inoculated into 100 mL potato dextrose broth medium (PDB; 200 g potatoes, 1 g KH_2_PO_4_, 1 g MgSO_4_·7H_2_0, 10 mg vitamin B1, and 20 g glucose, with a fixed volume of 1000 mL after preparation), shaking the fungus to form spherical mycelium with a diameter of about 4.0 mm at 25 °C at 200 rpm, in order to obtain a liquid strain. Then, 5 mL of the liquid strain were inoculated into the rice culture medium (30 g high-quality rice and 35 mL PDB medium were added to a 350 mL culture flask), cultured for 5 d at 25 °C under dark conditions, and then exposed to light at 20 °C for 2 days. The mycelium surface was scratched with inoculation rings to induce fruiting body buds. When the orange primordium was formed, the cap was replaced with a breathable cap to facilitate the development of fruiting bodies. During the continuous culture of fruiting bodies, two 4-W fluorescent lamps for continuous light were placed in the incubator with an illumination intensity of 450–600 Lux. The morphology of the fruiting body was observed and recorded.

### 2.6. Sample Treatment and Western Blot Analysis

Mycelium samples stored at –55 °C were ground into a powder in a mortar, frozen at –55 °C, and then added to a 2 mL centrifuge tube. In the centrifuge tube, 300 μL phosphate-buffered saline (PBS) (0.1 mol/L, pH 7.2, containing protease inhibitors for fungi and yeast) were pre-added, immediately shaken, and mixed, and the biomass concentration of the samples was adjusted to 1.0 g/mL. Samples were extracted overnight at 4 °C. After overnight extraction, centrifugation was performed at 12,000 rpm, and the supernatant was collected. If there was still suspended matter, it would be centrifuged again. After adding 5× denatured protein loading buffer and bathing at 100 °C for 5 min, the protein samples were loaded successively with a loading volume of 40 μL for each sample (32 μL sample protein + 8 μL loading buffer). The protein samples were subjected to SDS-PAGE (concentration gel 5%, separation gel 12%) at 120 V for 2.5 h, then electrotransferred to a PVDF membrane. The PVDF membrane was incubated in a closed solution (5% skim milk powder) for 2 h and then washed with PBST (pH 7.4 PBS added 0.1% Tween-20) 3 times, 10 min each time. The membrane was incubated in anti-CmFRQ2 at 4 °C overnight, washed with PBST 3 times, then incubated with sheep anti-rabbit IgG-HRP for 1 h, washed with PBST 3 times, and then a DAB staining kit was used for color development. The grayscale value of the protein bands was determined by the software ImageJ (version ImageJ 1.x, https://imagej.net/software/imagej/, accessed on 1 April 2023).

## 3. Results

### 3.1. Morphology of C. militaris Strain TN Colony Treated with Fungicides

*C. militaris* strain TN was inoculated onto PDA medium containing a certain concentration of fungicides, and colony morphology was observed on the 5th and 10th days (Figure 1). The results showed that the colonies treated with amphotericin B were the smallest on the 5th day of growth (Figure 1B). L-cysteine-treated colonies (Figure 1C), terbinafine-treated colonies (Figure 1D), and 5-fluorocytosine-treated colonies (Figure 1E) grew similarly to control colonies (Figure 1A). On the 10th day of growth, amphotericin B-treated colonies were also the smallest (Figure 1G), followed by terbinafine-treated colonies (Figure 1I). L-cysteine-treated colonies (Figure 1H) and 5-fluorocytosine-treated colonies (Figure 1J) were similar to the control colonies (Figure 1F), and both were in contact with the Petri plate wall. The mycelium in the colonies treated by the four fungicides was relatively tight (Figure 1B–E,G–J). These results indicate that amphotericin B inhibited the growth of the *C. militaris* strain TN mycelium more obviously, followed by terbinafine.

### 3.2. Morphology of C. militaris Strain CmFRQ454 Colony Treated with Fungicides

*C. militaris* strain CmFRQ454 was inoculated onto PDA medium containing a certain concentration of fungicides, and colony morphology was observed on the 5th and 10th days (Figure 2). The results showed that the colonies treated with terbinafine were the smallest on the 5th day of growth (Figure 2D). Amphotericin B-treated colonies (Figure 2B), L-cysteine-treated colonies (Figure 2C), and 5-fluorocytosine-treated colonies (Figure 2E) grew faster than control colonies (Figure 2A). On the 10th day of growth, terbinafine-treated colonies were also the smallest (Figure 2I), amphotericin B-treated colonies (Figure 2G), L-cysteine-treated colonies (Figure 2H), and 5-fluorocytosine-treated colonies (Figure 2J) were similar to the control colonies (Figure 2F). The mycelium in the colony treated with 5-fluorocytosine was more loose, which is beneficial for ventilation in the mycelium (Figure 2E,J). However, the mycelia in the colonies treated with other fungicides were relatively tight (Figure 2B–D,G–I). These results indicate that terbinafine inhibited the growth of *C. militaris* strain CmFRQ454 mycelium more obviously. In the early stage of colony culture, amphotericin B, L-cysteine, and 5-fluorocytosine could promote the *C. militaris* strain CmFRQ454 colony growth to some extent.

### 3.3. Analysis of CmFRQ in the Subculture S_0_ Mycelium of C. militaris Strain TN Treated with Fungicides

Previous studies had shown that the CmFRQ antibody Anti-FRQ2 prepared by us could detect the CmFRQ degradation forms of 50 kD and 33 kD proteins (to be published separately). When extracting mycelium protein, the content of mycelium was always fixed, and the total protein in 1 g of mycelium sample was extracted with 1 mL of PBS. Therefore, the relative content of CmFRQ in the mycelium was compared under the same biomass, as was the case in subsequent studies.

Western blot analysis was performed on the proteins extracted from the *C. militaris* strain TN mycelium treated with fungicides for 5 and 10 days. The detection of the antibody Anti-FRQ2 showed that the relative content of CmFRQ in S_0_-amphotericin B mycelium (mycelium of *C. militaris* strain TN treated with amphotericin B) (Figure 3A: 5-1) showed no significant change compared to the control group (Figure 3A: 5-CK) on the 5th day, and there was a significant increase compared to the control group (Figure 3A: 10-CK) on the 10th day (Figure 3A: 10-1). The relative content of CmFRQ in the *C. militaris* strain TN treated with L-cysteine was lower compared to that of the control group on the 5th day (Figure 3A: 5-CK and 5-2), and there was no significant change compared to the control group on the 10th day (Figure 3A: 10-CK and 10-2). On the 5th day of treatment with terbinafine, the relative content of CmFRQ in the *C. militaris* strain TN showed a certain increase compared to the control group (Figure 3B: 5-CK and 5-3), and on the 10th day, there was a sharp increase compared to the control group (Figure 3B: 10-CK and 10-3). On the 5th day, the relative content of CmFRQ in the *C. militaris* strain TN treated with 5-fluorocytosine showed a certain increase compared to the control group (Figure 3B: 5-CK and 5-4), and on the 10th day, there was a sharp increase compared to the control group (Figure 3B: 10-CK and 10-4). These results indicate that the treatment of amphotericin B, terbinafine, and 5-fluorocytosin could all increase the relative content of CmFRQ in the mycelium of *C. militaris* strain TN to a certain extent, especially in the late stage of culture close to the aging stage of the mycelium. L-cysteine had little effect on CmFRQ expression.

### 3.4. Analysis of CmFRQ in Subculture S_0_ Mycelium of C. militaris Strain CmFRQ454 Treated with Fungicides

The mycelium of *C. militaris* strain CmFRQ454 treated with amphotericin B showed no significant changes in the relative content of CmFRQ on the 5th day compared to the control group (Figure 4A: 5-CK and 5-1) but significantly decreased on the 10th day compared to the control group (Figure 4A: 10-CK and 10-1). The relative content of CmFRQ in the mycelium of *C. militaris* strain CmFRQ454 treated with L-cysteine decreased compared to the control group on the 5th and 10th days (Figure 4A: 5-CK, 5-2, 10-CK, and 10-2). The amount of CmFRQ in the mycelium of *C. militaris* strain CmFRQ454 treated with terbinafine decreased significantly compared to the control group on the 5th and 10th days (Figure 4B: 5-CK, 5-3, 10-CK, and 10-3). The mycelium of *C. militaris* strain CmFRQ454 was treated with 5-fluorocytosine, and the amount of CmFRQ significantly decreased on the 5th and 10th days (Figure 4B: 5-CK, 5-4, 10-CK, and 10-4). These results indicate that none of these four fungicides significantly increased the amount of CmFRQ in the *C. militaris* strain CmFRQ454 and even had a certain inhibitory effect.

### 3.5. Analysis of CmFRQ in Subculture S_1_ Mycelium of C. militaris Strain TN after Removal of Fungicide

The subculture S_0_ mycelium of the *C. militaris* strain TN treated with fungicides was transferred to the PDA medium without fungicides to obtain the subculture S_1_ mycelium without fungicide treatment. When the subculture S_1_ mycelium was cultured for 5 days, the relative content of CmFRQ in the S_1-_amphotericin B mycelium (Figure 5A: 5-1), S_1_-L-cysteine mycelium (Figure 5A: 5-2), S_1_-terbinafine mycelium (Figure 5A: 5-3), and S_1_-5-fluorocytosine mycelium (Figure 5A: 5-4) compared to that in the control group (Figure 5A: 5-CK) showed no significant changes. However, when the subculture S_1_ mycelia was cultured for 10 days, the relative content of CmFRQ in the S_1_-amphotericin B mycelium (Figure 5A: 10-1), S_1_-L-cysteine (Figure 5A: 10-2), S_1_-terbinafine (Figure 5A: 10-3), and S_1_-5-fluorocytosine (Figure 5A: 10-4) compared to that in the control group (Figure 5A: 10-CK) increased significantly. This indicates that in subculture S_1_ mycelium treated with the removal of these four fungicides, the early circadian clock in the *C. militaris* strain TN mycelium did not change much, but the late circadian clock changed greatly.

### 3.6. Analysis of CmFRQ in Subculture S_1_ Mycelium of C. militaris Strain CmFRQ454 after Removal of Fungicide

Subculture S_0_ mycelium of *C. militaris* strain CmFRQ454 treated with fungicides was transferred to the culture medium without fungicide to obtain subculture S_1_ mycelium without fungicide treatment. When subculture S_1_ mycelium was cultured for 5 days, the relative content of CmFRQ in S_1_-amphotericin B (Figure 6A: 5-1), S_1_-L-cysteine (Figure 6A: 5-2), S_1_-Terbinefin (Figure 6A: 5-3) and S_1_-5-fluorocytosine (Figure 6A: 5-4) mycelium compared with the control (Figure 6A: 5-CK) was decreased, and that in the S_1_-L-cysteine was more decreased (Figure 6A: 5-2). When subculture S_1_ mycelia was cultured for 10 days, the relative content of CmFRQ in S_1_-amphotericin B mycelium (Figure 6A: 10-1), S_1_-L-cysteine (Figure 6A: 10-2), S_1_-terbinafine (Figure 6A: 10-3) and S_1_-5-fluorocytosine (Figure 6A: 10-4) mycelium compared with the control group (Figure 6A: 10-CK) was decreased significantly. These results indicated that the expression of CmFRQ in subculture S_1_ mycelium after removal of these four fungicides is down-regulated, and the operation of the circadian clock may be weakened.

### 3.7. Morphological Analysis of Subculture S_1_ Fruiting Bodies of C. militaris Strain TN after Removal of Fungicides

The subculture S_1_ mycelium of *C. militaris* strain TN was inoculated in a culture bottle for fruiting body culture, and the fruiting body morphology was observed after 40 days of culture (Figure 7). Compared with the control group (Figure 7A), the differences appeared in the fruiting bodies of subculture S1 despite the absence of fungicide effect. The results of S_1_-amphotericin B (Figure 7B), S_1_-L-cysteine (Figure 7C) and S_1_- terbinafine (Figure 7D) cultures showed certain degenerated, shorter, and more deformed fruiting body than the control group (Figure 7A). The appearance of albino mycelium on the stroma obtained from S_1_-L-cysteine (Figure 7C) and S_1_-terbinafine (Figure 7D) cultures affected the normal growth of the fruiting body. The fruiting bodies obtained by S_1_-5-fluorocytosine culture were thinner than those in the control group, but no malformed fruiting body and albino mycelium was present on the stroma (Figure 7E). These results indicated that the effect of fungicide on *C. militaris* strain TN could continue to the next generation subculture without fungicide treatment, resulting in worse fruiting body growth. However, the fruiting body obtained after 5-fluorocytosine treatment showed relatively good performance, with almost no degeneration.

### 3.8. Morphological Analysis of Subculture S_1_ Fruiting Bodies of C. militaris Strain CmFRQ454 after Removal of Fungicide

The *C. militaris* strain CmFRQ454 was once a rejuvenated strain, and after multiple generations of asexual transfer culture, the fruiting body showed degeneration (Figure 8A). The subculture S_1_ mycelium of *C. militaris* strain CmFRQ454 was inoculated in a culture bottle for fruiting body culture, and the morphology of fruiting body was observed after 40 days of culture (Figure 8). Compared with the control group (Figure 8A), the fruiting bodies obtained from S_1_ amphotericin B (Figure 8B), S_1_-L-cysteine (Figure 8C), and S_1_-terbinafine (Figure 8D) cultures were still degenerated, similar to the control group (Figure 8A), showing deformed enlargement of fruiting bodies, bifurcation, shorter than normal fruiting bodies, and albino mycelium on the stroma. Among them, the degeneration of fruiting bodies obtained from S_1_- terbinafine culture was more obvious (Figure 8D), and the number of fruiting bodies decreased and were shorter than that of the control group (Figure 8A). However, the fruiting body obtained by S_1_-5-fluorocytosine culture showed rejuvenation, normal growth, few deformities, and longer than control fruiting body (Figure 8E). These results indicated that the treatment of *C. militaris* strain CmFRQ454 with fungicides had different effects on fruiting bodies, which could lead to further degeneration of fruiting bodies, and also lead to the rejuvenation of the fruiting body.

## 4. Discussion

At present, research on the relationship between fungicides and fungal diversity mainly focuses on the relationship between the resistance of fungicides and the genetic diversity of fungi [1,4,5,6,52]. The main idea of our research was to investigate the changes in the circadian clock and fruiting body of *C. militaris* under the influence of fungicides. After treating the mycelium of *C. militaris* with four fungicides at a certain concentration, the mycelium was not completely killed, but a certain non-lethal selection pressure was given. Under the action of four fungicides, the clock protein CmFRQ, mycelial growth, and fruiting body development of *C. militaris* were all affected; in particular, the impact on fruiting bodies was bidirectional, which can lead to the degeneration of fruiting bodies and also promote their rejuvenation.

In this study, two strains of *C. militaris* with different mating types, TN and CmFRQ454, were used. Obviously, the genetic characteristics of these two strains were different. Therefore, the expression of CmFRQ in subculture S_0_ mycelium treated with four fungicides was different between the two strains. Three fungicides, namely amphotericin B, terbinafine, and 5-flucytocytosin, could promote an increase in the relative content of CmFRQ in the mycelium of *C. militaris* strain TN, and this promotion effect could increase with time. However, high concentrations of L-cysteine inhibited the expression of CmFRQ in *C. militaris* strain TN mycelium. These four fungicides could reduce the relative content of CmFRQ in strain CmFRQ454 mycelium, and 5-fluorocytosin promoted the highest degree of CmFRQ reduction. These results indicate that the circadian clock of *C. militaris* with different mating types was disturbed differently during the stress of fungicides on the mycelium of *C. militaris*.

After the stress of the fungicide was removed, the effect of the fungicide on the clock protein CmFRQ in mycelium was still retained in the subculture S_1_ obtained from the S_0_ transferring culture. The results showed that the relative expression of CmFRQ was increased in the mycelium of subculture S_1_ of strain TN treated with amphotericin B, L-cysteine, terbinafine, and 5-fluorocytosine. However, in the mycelium of subculture S_1_ of *C. militaris* strain CmFRQ454, the situation was exactly the opposite, with a relative decrease in CmFRQ expression. After removing the stress of fungicides, the effect of the stress will continue, and the response to different mating types of *C. militaris* is exactly the opposite.

After treating the *C. militaris* strains TN and CmFRQ454 with these four fungicides, the mycelium in the colonies was relatively tight, but only the mycelium in the colonies of strain CmFRQ454 treated with 5-fluorocytosine was relatively loose, which was conducive to the formation of fruiting bodies. Most of the fruiting bodies obtained from subculture S_1_ after removing the fungicides showed the morphological degeneration. However, 5-fluorocytosine led to the revival of the degenerated *C. militaris* strain CmFRQ454. It is known that 5-fluorocytosine interferes with RNA transcription, thereby affecting protein synthesis [45,46]. However, from the performance of these two strains, 5-fluorocytosine promotes the synthesis of CmFRQ in *C. militaris* strain TN while inhibiting it in *C. militaris* strain CmFRQ454. Therefore, the expression of circadian clock proteins could correspond to the degeneration or rejuvenation of the fruiting body. There are few studies on the bidirectional effects of fungicides on fungal growth and reproduction. Some studies have shown that DMI fungicides could promote the spore production of *Colletotrichum* spp. at low concentrations [53]. Obviously, the bidirectional action of fungicides brings more diversity to fungal physiology and morphology.

*C. militaris* is a fungus prone to degeneration, especially during subculture [50]. Some researchers have regarded the degeneration of fungi as an aging problem and determined that the mechanism of degeneration is the same as that of aging, and high levels of reactive oxygen species (ROS) lead to molecular damage and degeneration [54,55,56,57]. Studies have shown that ROS can regulate the important components of the circadian oscillator in *N. crassa*, and it has also been determined that ROS homeostasis is controlled by the circadian clock, and ROS levels exhibit a circadian rhythm [58,59,60,61,62]. The mycelium of the degenerated *C. militaris* strain was also found to contain more ROS and higher levels of oxidative stress, with a significant negative correlation between its oxidative stress and strain degeneration [63]. Therefore, we speculate that the effect of fungicides on *C. militaris* could affect the balance of ROS in *C. militaris* cells, thereby affecting the degeneration of *C. militaris*, which is closely related to its circadian clock.

## 5. Conclusions

Four fungicides, amphotericin B, L-cysteine, terbinafine, and 5-fluorocytosine, were used to treat two *C. militaris* strains, strain TN and strain CmFRQ454. Treatment with the four fungicides resulted in changes in the expression of the clock protein CmFRQ and the fruiting body morphology of the two strains.

Amphotericin B, terbinafine, and 5-fluorocytosine could increase the relative content of CmFRQ in the subculture S_0_ mycelium of *C. militaris* strain TN, while high concentrations of L-cysteine reduced the relative content of CmFRQ in the subculture S_0_ mycelium of *C. militaris* strain TN. These four fungicides could reduce the relative content of CmFRQ in the subculture S_0_ mycelium of *C. militaris* strain CmFRQ454. After removing the stress of fungicides, the effect of the stress of fungicides would continue. The relative content of CmFRQ was increased in the subculture S_1_ mycelium of strain TN treated with amphotericin B, L-cysteine, terbinafine, and 5-fluorocytosine, but the opposite result appeared in the subculture S_1_ mycelium of *C. militaris* strain CmFRQ454.

The fruiting bodies of subculture S_1_ of strain TN or strain CmFRQ454 still showed degeneration after removing the treatment of four fungicides, but rejuvenation was provided to the fruiting bodies of subculture S_1_ of strain CmFRQ454 after removing the treatment of 5-fluorocytosine.

## Figures and Tables

**Figure 1 jof-10-00150-f001:**
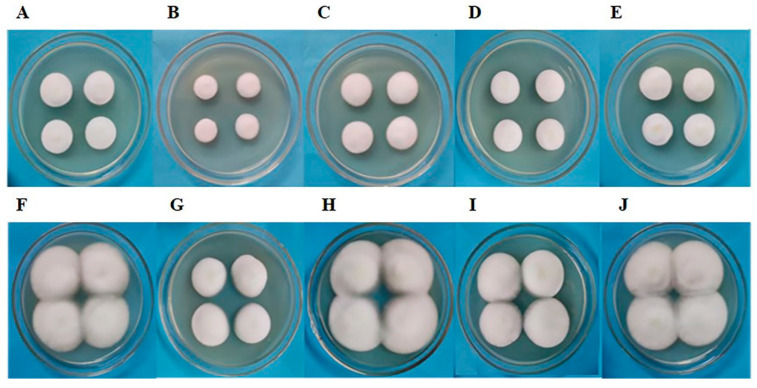
The colonies of *C. militaris* strain TN treated with amphotericin B (**B**), L-cysteine (**C**), terbinafine (**D**), and 5-fluorocytosine (**E**) were cultured for 5 days, and the colonies of strain TN treated with amphotericin B (**G**), L-cysteine (**H**), terbinafine (**I**), and 5-fluorocytosine (**J**) were cultured for 10 days. At the same time, the control colonies of *C. militaris* TN were also cultured for 5 days (**A**) and 10 days (**F**), respectively.

**Figure 2 jof-10-00150-f002:**
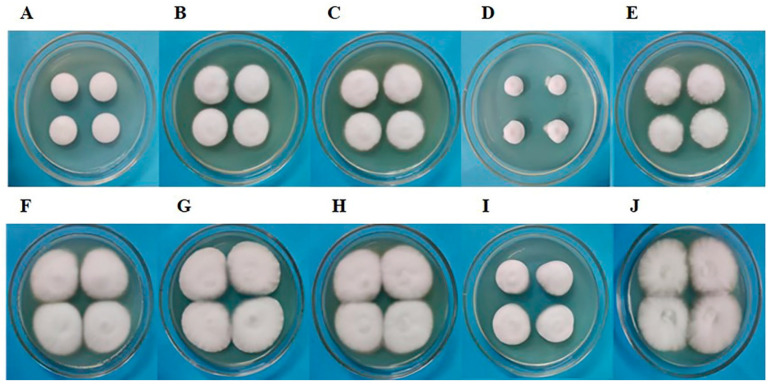
The colonies of *C. militaris* strain CmFRQ454 treated with amphotericin B (**B**), L-cysteine (**C**), terbinafine (**D**), and 5-fluorocytosine (**E**) were cultured for 5 days, and the colonies of strain CmFRQ454 treated with amphotericin B (**G**), L-cysteine (**H**), terbinafine (**I**), and 5-fluorocytosine (**J**) were cultured for 10 days. At the same time, the control colonies of *C. militaris* CmFRQ454 were also cultured for 5 days (**A**) and 10 days (**F**), respectively.

**Figure 3 jof-10-00150-f003:**
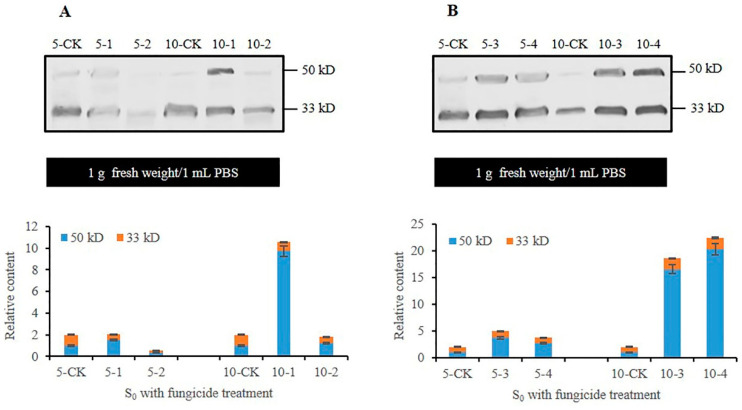
The relative content of CmFRQ in *C. militaris* strain TN treated with different fungicides was analyzed by Western blotting. (**A**) The Western blotting analysis of CmFRQ in *C. militaris* strain TN treated with amphotericin B (A: 5-1 and 10-1) and L-cysteine (A: 5-2 and 10-2) was implemented (*p* < 0.05); (**B**) The Western blotting analysis of CmFRQ in *C. militaris* strain TN treated with terbinafine (B: 5-3 and 10-3) and 5-fluorocytosine (B: 5-4 and 10-4) was implemented (*p* < 0.05). Note: Total protein was extracted from 0.1 g mycelium samples with 0.1 mL PBS, i.e., 1.0 g fresh weight/1 mL PBS.

**Figure 4 jof-10-00150-f004:**
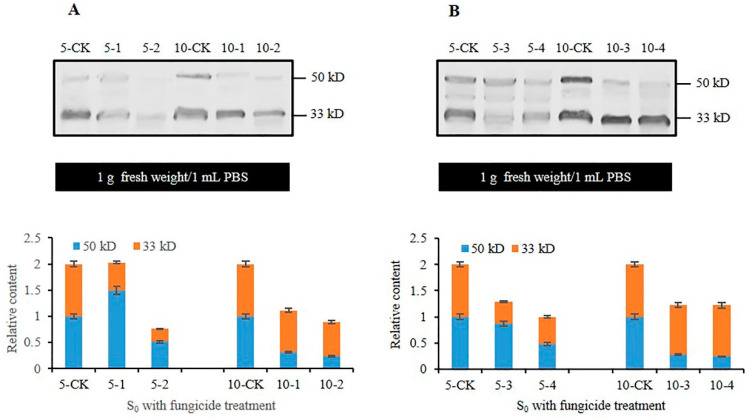
The relative content of CmFRQ in *C. militaris* strain CmFRQ454 treated with different fungicides was analyzed by Western blotting. (**A**) The Western blotting analysis of CmFRQ in *C. militaris* strain CmFRQ454 mycelium treated with amphotericin B (A: 5-1 and A: 10-1) and L-cysteine (A: 5-2 and A: 10-2) was implemented (*p* < 0.05); (**B**) The Western blotting analysis of CmFRQ in *C. militaris* strain CmFRQ454 mycelium treated with terbinafine (B: 5-3 and B: 10-3) and 5-fluorocytosine (B: 5-4 and B: 10-4) was implemented (*p* < 0.05). Note: Total protein was extracted from 0.1 g mycelium samples with 0.1 mL PBS, i.e., 1.0 g fresh weight/1 mL PBS.

**Figure 5 jof-10-00150-f005:**
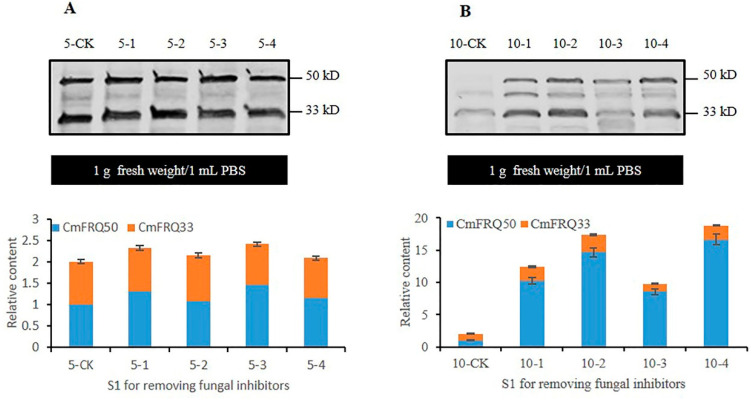
The relative content of CmFRQ in S_1_ mycelium of *C. militaris* strain TN was analyzed by Western blotting. (**A**) The Western blotting analysis of CmFRQ in the S_1-_amphotericin B (from S_0_ with amphotericin B treatment) (A: 5-1 and 10-1), S_1_-L-cysteine (from S_0_ with L-cysteine treatment) (A: 5-2 and 10-2) was implemented (*p* < 0.05); (**B**) The western blotting analysis of CmFRQ in the S_1_- terbinafine (from S_0_ with terbinafine treatment) (A: 5-3 and 10-3) and S_1_-5-fluorocytosine (from S_0_ with 5-fluorocytosine treatment) (A: 5-4 and 10-4) mycelium are was implemented (*p* < 0.05). Note: Total protein was extracted from 0.1 g mycelium samples with 0.1 mL PBS, i.e., 1.0 g fresh weight/1 mL PBS.

**Figure 6 jof-10-00150-f006:**
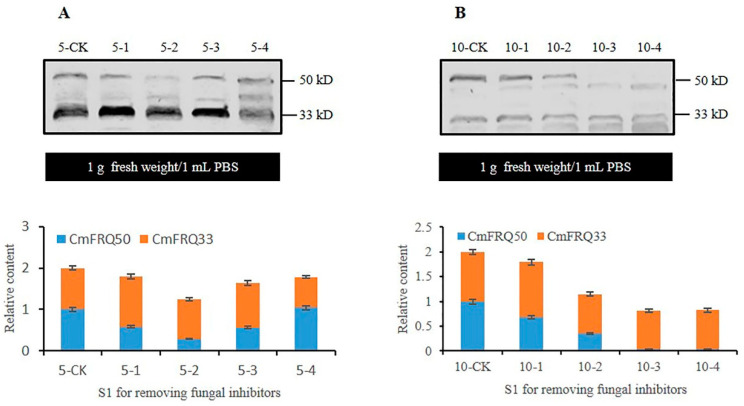
The relative content of CmFRQ in S_1_ mycelium of *C. militaris* strain CmFRQ454 was analyzed by Western blotting. (**A**) The Western blotting analysis of CmFRQ in the S_1-_amphotericin B (from S_0_ with amphotericin B treatment) (A: 5-1 and 10-1) and S_1_-L-cysteine (from S_0_ with L-cysteine treatment) (A: 5-2 and 10-2) was implemented (*p* < 0.05); (**B**) The Western blotting analysis of CmFRQ in the S_1_- terbinafine (from S_0_ with terbinafine treatment) (A: 5-3 and 10-3) and S_1_-5-fluorocytosine (from S_0_ with 5-fluorocytosine treatment) (A: 5-4 and 10-4) mycelium was implemented (*p* < 0.05). Note: Total protein was extracted from 0.1 g mycelium samples with 0.1 mL PBS, i.e., 1.0 g fresh weight/1 mL PBS.

**Figure 7 jof-10-00150-f007:**
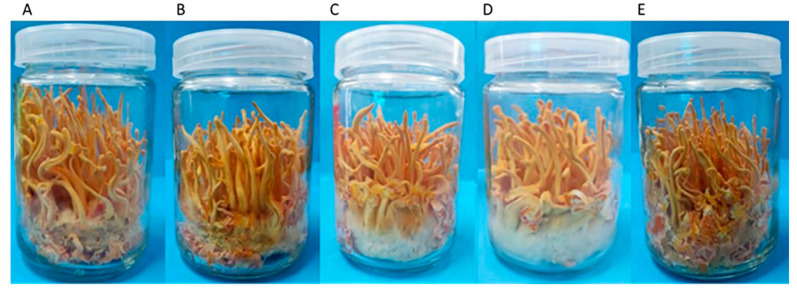
The fruiting body of subculture S_1_ of *C. militaris* strain TN after removal of fungicide. The fruiting bodies obtained by S_1_-amphotericin B, S_1_-L-cysteine, S_1_-terbinafine and S_1_-5-fluorocytosine culture, respectively, are shown in (**B**–**E**). The control fruiting body was shown in (**A**).

**Figure 8 jof-10-00150-f008:**
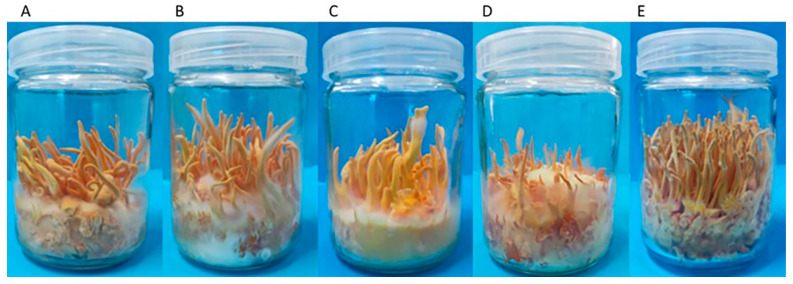
The fruiting body of subculture S_1_ of *C. militaris* strain CmFRQ454 after removal of fungicide. The fruiting bodies obtained by S_1_-amphotericin B, S_1_-L-cysteine, S_1_-terbinafine and S_1_-5-fluorocytosine culture, respectively, are shown in (**B**–**E**). The control fruiting body is shown in (**A**).

## Data Availability

Data are contained within the article.

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
