# Peer review of "The Stress of Fungicides Changes the Expression of Clock Protein CmFRQ and the Morphology of Fruiting Bodies of Cordyceps militaris"

_jof, 2024, doi:10.3390/jof10020150_

Round 1

Reviewer 1 Report

Comments and Suggestions for Authors

Journal of Fungi - 2814888

Recommendation: Publish in Journal of Fungi after minor revision

In my opinion, a very thorough and useful study has been carried out by the authors. Work is focused on a very important problem. The study has been fully completed. All necessary data are provided. In order to improve the quality of the article, I suggest a few comments.

Minor points:

1.         Authors should include more recent updates on the topic and compare how this study advances current knowledge in the Introduction section.

2.         I recommend that the "Conclusion" section be added.

Author Response

Dear Reviewer

We have finished to revise the article according to your comments. The attachments are the revised paper, and below is a list of responses to the comments. If you have any questions, notify us please.

Thank you very much.

Best regards,

Ming-Jia Fu

Responses to minor points:

  1. Authors should include more recent updates on the topic and compare how this study advances current knowledge in the Introduction section.

Response: In the "Introduction" section, we have supplemented and updated some contents.

  1. I recommend that the "Conclusion" section be added.

Response: In the paper, we have added the "Conclusion" section.

Reviewer 3 Report

Comments and Suggestions for Authors

Dear Authors,

very interesting idea of research. Please make slight changes in the text. See comments in pdf file.

Author Response

Dear Reviewer

We have finished to revise the article according to your comments. The attachments are the revised paper, and below is a list of responses to the comments. If you have any questions, notify us please.

Thank you very much.

Best regards,

Ming-Jia Fu

Responses to Reviewers' comments

Line 19: do not start sentence with But....please change

Response: revised.

Line 53: change The in the

Response: revised.

Line 108: what was the diameter of the hole?

Response: revised. Diameter is 6 mm

Line 123: by using.....which unit?

Response: Here we don't understand the meaning of Reviewers, but we have checked and modified all units in the section 2.3, 2.4 and 2.5.
